# Interlaboratory Validation of Toxicity Testing Using the Duckweed *Lemna minor* Root-Regrowth Test

**DOI:** 10.3390/biology11010037

**Published:** 2021-12-27

**Authors:** Jihae Park, Eun-Jin Yoo, Kisik Shin, Stephen Depuydt, Wei Li, Klaus-J. Appenroth, Adam D. Lillicrap, Li Xie, Hojun Lee, Geehyoung Kim, Jonas De Saeger, Soyeon Choi, Geonhee Kim, Murray T. Brown, Taejun Han

**Affiliations:** 1Laboratory of Plant Growth Analysis, Ghent University Global Campus, 119-5, Songdomunhwa-ro, Incheon 21985, Korea; Jihae.park@ghent.ac.kr (J.P.); Stephen.Depuydt@ghent.ac.kr (S.D.); hojun.lee@ugent.be (H.L.); Jonas.desaeger@ghent.ac.kr (J.D.S.); 2Environmental Measurement & Analysis Center, Department of Environmental Infrastructure Research, National Institute of Environmental Research (NIER), 42, Hwangyeong-ro, Incheon 22689, Korea; ejyoo@korea.kr; 3Water Environmental Engineering Research Division, National Institute of Environmental Research (NIER), 42, Hwangyeong-ro, Incheon 22689, Korea; envi95@korea.kr; 4Laboratory of Aquatic Plant Biology, Wuhan Botanical Garden, Chinese Academy of Sciences, Wuhan 430074, China; liwei@wbgcas.cn; 5Matthias Schleiden Institute, Plant Physiology, Friedrich Schiller University Jena, Dornburger Str. 159, 07743 Jena, Germany; klaus.appenroth@uni-jena.de; 6Norwegian Institute for Water Research (NIVA), Økernveien 94, NO-0579 Oslo, Norway; Adam.Lillicrap@niva.no (A.D.L.); Li.Xie@niva.no (L.X.); 7Environmental Technology Center, Environmental Corporation of Incheon, 6, Songdogukje-daero 372, Incheon 22014, Korea; water6904@naver.com; 8Department of Marine Science, Incheon National University, 119, Academy-ro, Incheon 22012, Korea; chlthdus0501@hanmail.net (S.C.); jkh-011@hanmail.net (G.K.); 9School of Biological and Marine Sciences, University of Plymouth, Plymouth, Devon PL4 8AA, UK; m.t.brown@plymouth.ac.uk; 10Department of Animal Sciences and Aquatic Ecology, Ghent University, Coupure Links 653-block F, B-9000 Gent, Belgium

**Keywords:** interlaboratory comparison, internationalized methods, *Lemna minor*, metals, root regrowth

## Abstract

**Simple Summary:**

Duckweed (*Lemna minor*) is commonly used as a phytotoxicity test organism, adopted by the main international standardization organizations (ISO, OECD, USEPA, ASTM). For duckweed tests, measurements of fronds or biomass are usually preferred with a standard exposure period of at least 7 days. The proposed root- regrowth test differs from other internationally standardized methods in several important aspects: (a) the test can be performed within 72 h; (b) the test vessel was a 24-well cell plate; (c) the required volume of test water samples was 3 mL; (d) roots were excised before exposure and newly developed roots then measured. The validation of the new test method by interlaboratory comparison tests confirmed that the *Lemna* root bioassay is valid and reliable. The root growth test is therefore a valuable tool for rapid toxicity screening of wastewater effluents and hazardous pollutants in natural waters because it is simple to perform, quick to conduct, cost-effective to operate, and can have operational benefits for testing time, since management decisions need to be made promptly in the event of unpredictable pollution events.

**Abstract:**

The common duckweed (*Lemna minor*), a freshwater monocot that floats on the surfaces of slow-moving streams and ponds, is commonly used in toxicity testing. The novel *Lemna* root- regrowth test is a toxicity test performed in replicate test vessels (24-well plates), each containing 3 mL test solution and a 2–3 frond colony. Prior to exposure, roots are excised from the plant, and newly developed roots are measured after 3 days of regrowth. Compared to the three internationally standardized methods, this bioassay is faster (72 h), simpler, more convenient (requiring only a 3-mL) and cheaper. The sensitivity of root regrowth to 3,5-dichlorophenol was statistically the same as using the conventional ISO test method. The results of interlaboratory comparison tests conducted by 10 international institutes showed 21.3% repeatability and 27.2% reproducibility for CuSO_4_ and 21.28% repeatability and 18.6% reproducibility for wastewater. These validity criteria are well within the generally accepted levels of <30% to 40%, confirming that this test method is acceptable as a standardized biological test and can be used as a regulatory tool. The *Lemna* root regrowth test complements the lengthier conventional protocols and is suitable for rapid screening of wastewater and priority substances spikes in natural waters.

## 1. Introduction

Assays using sentinel organisms are often employed to assess pollutant-induced ecological risks. Given the wide biodiversity of organisms in the world, with close to 6.5 million species on land and another 2.2 million species in the oceans [1], it is impossible to fully elucidate the potential ecotoxicological effects of chemicals on all organisms. Therefore, ecotoxicologists develop standardized toxicity testing methods for a small set of indicator species, with the choice of model organisms largely dependent on their relative sensitivities to specific pollutants, relevance and ease of use. Several international bodies regulate toxicity test standardization, including the International Organization of Standardization (ISO) [2], the Organization for Economic Cooperation and Development (OECD) [3], the US Environmental Protection Agency (US EPA) [4] and the American Society for Testing and Materials (ASTM) [5]. To date, 124 aquatic bioassay standards have been registered by these organizations, 21, 90 and 13 of which use producer, consumer and decomposer species, respectively (Appendix A).

The aquatic plants of the Lemnaceae family are attractive experimental model organisms due to their simple structure, small size, degree of homogeneity, ease of culture and rapid growth (1.2- to 4.2-day doubling time) [6,7,8,9,10]. Duckweeds are a major group of primary producers at the base of trophic hierarchies in aquatic ecosystems. These plants are of prime importance, since any negative impacts on duckweed can have serious consequences higher up the food chain, leading to changes in the diversity and functionality of whole aquatic ecosystems. In particular, species of the genus *Lemna* are powerful test organisms due to their wide geographical distribution and their key roles in primary production, nutrient cycling and the structuring of aquatic ecosystems by providing food and protective environments (e.g., nurseries and habitats) for other organisms [11,12]. As of 2015, at least 2120 studies have been published on the effects of phytotoxins on duckweeds [13]. For these reasons, laboratory toxicity testing using *Lemna* spp. (duckweed) is now a choice methodology for assessing impacts on freshwater systems [14,15]. 

Since the 1930s duckweeds have been used to assess the effects of a wide range of contaminants, including herbicides, pesticides, fertilizers and various other inorganic and organic compounds in 1979, *L. minor* was proposed as a ‘representative’ aquatic macrophyte for assessing the environmental safety of chemicals. Since then it has been widely used in phytotoxicity testing [16] and several standardized methodologies have been adopted by the major international standardization agencies, for example: in the USA, *L. minor* test methods were published by the ASTM [17] and the USEPA [18]; in Canada, Environment Canada (EC) published a biological test method for measuring growth inhibition using *L. minor* [19], in Europe, duckweed test standards were published by the Association Française de Normalisation (AFNOR) [20], the Swedish Standards Institute (SIS) [21], the ISO [22] and the OECD [23].

Various literature reviews have been published that help synthesize the vast body of valuable information on toxicity testing with duckweeds. For example, Wang [24] provided on overview of test results using *L. minor*, *Lemna valdiviana*, *Lemna polyrrhiza*, *Lemna gibba*, *Lemna perpusilla* and *Spirodela oligorhiza* for single compounds including Ba, Cd, Cl, Cr(VI), Cu, Pb, Mn, Ni, Se, SO_4_, Zn, alcohol ethoxylate, alachlor, atrazine, carbofuran, o-cresol, cetyl trimethyl ammonium chloride, di-(2-ethylhexyl)phthalate, ethylene glycol, C11,8 linear alkylbenzene sulfonate, sodium dodecyl sulfate, 2,4,6-trichlorophenol, glyphosate, pentachlorophenol, phenol and salicylic acid and for complex mixtures of these chemicals. In the review by Baudo et al. [13] they presented comparative results for two duckweed genera, *Spirodela* and *Lemna*, exposed to herbicides (thinfenylsulfuron-methyl, tribenuron-methyl, metribuzin, lenacil, tritosulfuron, linuron, terbutylazine, imazamox, metamitron), inorganic and organic compounds (3,5-dichlorophenol, acetone, KCl, ethanol) and metals (Ag, Cu, Cd, Ni, Hg, Co, Cr(VI), Zn and Ziegler et al. [16] reviewed the standardized toxicity testing protocols and provided information on the toxicity of a range of metals (As, Cd, Cr, Co, Cu, Hg, Ni, Ag, TL, Zn) to *L. minor* and herbicides (chlorsulfuron, alachlor, isoproturon, paraquat, 4,6-dintro-o-cresol, chlorpropham, gulufosinate, S-ethyl dipropyl carbamothiolate, naphtalam, glyphosate) to *Lemna aequinoctialis*. It is evident that since the turn of the current century, duckweeds, one of the “fastest-growing angiosperms”, have been gaining increasing attention in ecotoxicological research. In the journal of the International Steering Committee on Duckweed Research and Applications (ISCDRA), Edelman [25] reported that since the year 2000, more than 300 studies on the toxicity of chemicals or wastewaters to duckweeds have been published. It is therefore not surprising that toxicity tests using duckweeds have been integrated into environmental legislation and guidelines.

Standard *Lemna* bioassays are relatively straightforward and can be carried out in replicate test vessels containing >100 mL of test solution and two colonies of plants at the three-frond stage. These plants can easily be cultured in test vessels containing a specific nutrient medium by incubation at 25 °C with continuous illumination (usually 100 μmol m^–2^ s^–1^ photon flux density (PFD) light) in the laboratory. The standard exposure duration is 7 d (longer test periods increase interference from contamination), after which the inhibitory effects can be determined by measuring various endpoints, such as the number and size of fronds, wet or dry biomass or chlorophyll content [22]. 

While current standard *L. minor*-bioassays generate reproducible results and are cost-effective, a more rapid assessment of toxicity without loss of reproducibility and sensitivity would be a valuable development of existing protocols. A recent evaluation of the ecotoxicological significance of root growth as an endpoint has revealed that it is sensitive, precise and ecologically significant compared to more traditional endpoints e.g., frond growth or biomass [26,27]. Gopalapillai et al. [27] identified the average root length of *L. minor* as an optimal endpoint for the biomonitoring of mining wastewater for three reasons: accuracy (toxicological sensitivity to the pollutant), precision (lowest variance) and ecological relevance (direct exposure to metal contaminated wastewater). Subsequently, Park et al. [26] established a well-defined method to measure toxicity-concentration-dependent inhibition of root regrowth in three *Lemna* species. This protocol has several operational benefits over the more conventional ISO 20079 procedure, including its 72-h test duration, the requirement for only 3.0 mL test solution and the use of non-axenic plant material. In addition to standardization, good reproducibility and acceptable interlaboratory variability are two other important requirements when using bioassays as regulatory tools. Currently, no such information is currently available on the extent to which different laboratories reach the same conclusion using this bioassay or on identifying, any potential sources of variability between laboratories.

Here, we present a detailed protocol used to test the toxicity of contaminated freshwater samples using the *L. minor* root regrowth bioassay and an outline of how to analyze the data in a standardized manner. This procedure is detailed in a New Work Item Proposal (NWIP) for the International Organization for Standardization (ISO) currently under development as a protocol entitled ‘Aquatic toxicity test based on root regrowth in *Lemna minor* (ISO/AWI 4979)’ [28]. In addition, we describe (1) a comparison of the sensitivity to detect a standard toxicant (CuSO_4_) and a wastewater sample between the newly developed test and the traditional international standard methodology, and (2) the validation of the new test method by interlaboratory comparison tests.

## 2. Materials and Methods

### 2.1. Plant Material and Culture Conditions

The standard test organism in this test protocol is the duckweed *L. minor*, a freshwater floating plant (Figure 1). *Lemna minor* (CPCC 490), International Clone ID 5631 (Rutgers Duckweed Stock Cooperative, New Brunswick, NJ, USA) was obtained from the Canadian Phycological Culture Centre.

This duckweed stock culture was maintained in the laboratory at 25 ± 2 °C under 30–40 μmol photons m^–2^ s^–1^ of continuous light provided by a constant cool-white fluorescent light (FL 20 SS/18D, Philips Co., Bangkok, Thailand). The cultures were maintained in polypropylene containers (103 × 78.6 mm) containing Steinberg growth medium [29]. Distilled water was used to dilute the liquid medium, after which the pH was adjusted to 6.9 ± 0.2. The growth medium was replaced weekly.

### 2.2. Toxicity Testing Procedure

#### 2.2.1. Preparation of Medium

The necessary reagents and respective quantities required to prepare 1 L of Steinberg medium were outlined previously. Stock solutions of reagent-grade chemicals were added to 938 mL distilled water. The medium was stirred until all chemicals had dissolved. The pH of the liquid medium was adjusted to 6.9 ± 0.2 using either HCl or NaOH at ≤ 1 M after adding distilled water to each stock solution.

#### 2.2.2. Preparation of Test Solution and Test Dilutions

The test toxicant stock solution (1000 mg L^−1^ of CuSO_4_, CAS No. 7758-98-7, Showa Chem., Tokyo, Japan) was stored under cool, dry conditions until the test solutions were prepared. The test dilutions were prepared in volumetric flasks and distributed into the replicate test vessels, which were then left at room temperature for 1 h to allow the medium and the toxicant to equilibrate.

Different concentrations of the test substance were prepared using various dilutions of stock solutions with test medium. Range-finding tests were performed to determine the range of concentrations to be tested in the definitive test, initially using a wide range of concentrations (≥an order of magnitude). All experiments required a negative control involving an identical culture medium, test condition and procedure but excluding the test substance. 

A wastewater sample was collected from a wastewater treatment plant (°N 35.982, °W 129.50) and stored at 4 °C until needed. The wastewater toxicity test was conducted using a concentration series of 100%, 50%, 25%, 12.5% and 6.25% (*v/v*) of the original effluent water with test medium. The pH of all test samples was adjusted to approximately the optimal value for testing.

#### 2.2.3. Transfer of Test Organisms

Healthy frond colonies of duckweed that were dark green and consisted of two or three identical leaves attached were selected for experimentation. The roots of the selected fronds were excised using stainless scissors (Figure 2c). Individual rootless plants were then placed in each cell of 24-well plates by using tweezers (Figure 2d). Care was taken to ensure that the plants did not adhere to the side of the well. The transfer of *Lemna* to test solutions was done randomly across replicates within a concentration. The plate was covered and sealed with sealing tape to avoid evaporation of medium or test solution.

For the period of the experiment, the test organisms were cultured under defined conditions of 25 °C with continuous white light of 90–100 μmol m^−2^ s^−1^ (Figure 2e). The duration of the *Lemna* root growth inhibition test was 72 h. Determining the exposure time has been described by Park [30]. The length of the exposure period is equally important when assessing any toxic effects. Exposure durations from 2 to 7-d were tested using the lengths of regrown roots as the endpoint. In general, sensitivity decreased as exposure time increased, but there was an overlap in the 95% CIs between the 3-d and 6-d exposure period. Since rapid response times are often required to deal with chemical pollution events a short duration period was considered desirable; therefore, based on our findings, a 3-d fixed time period for the *L. minor* root regrowth test was chosen.

#### 2.2.4. Measurement Methods

After 72 h, fronds were transferred using tweezers onto a glass microscope slides, with the upper part of the frond adhering to the glass (Figure 2f,g). As the fronds were wet, the new roots could be easily straightened by gently manipulating them with tweezers. The distance between the camera and the glass slide was adjusted and fixed. Images of regrown roots were captured (Figure 2h) using ImageJ (NIH Bethesda, MD, USA). The length of the longest root of each plant was measured.

### 2.3. Time Required for Each Step

The preparation of the stock solution and culture medium took 1 h. Frond selection and root cutting also took 1 h (where n = 72). Transferring the sample plants into new or test medium took 15 min. The incubation period for the tests was 72 h. Once a test was completed, the processing times for harvesting and measurement of all sample plants was 30 min.

### 2.4. Troubleshooting

The fronds were sub-cultured at 25 °C under 24 h of continuous white light at an irradiance of 30–40 μmol m^–2^ s^–1^. The medium was replaced every 7 d, and the stock culture was maintained continuously unless uncontrolled contamination occurred. Cloudy medium in a *Lemna* stock culture indicated bacterial contamination, whereas contamination with mold was not evident until large colonies appeared in the medium or a slime layer developed in the vessel. Contaminated *Lemna* cultures were discarded.

Care was taken to ensure that the plants did not adhere to the sides of the wells. The transfer of fronds into test solutions was performed in random order across replicates within a concentration. The plates were covered and sealed with sealing tape to prevent the evaporation of the medium or test solution.

A static-type test was used to avoid changing the test solutions during the exposure period. When carrying out the experimental procedures, a fully randomized design was used to account for any variability in environmental conditions within the culture cabinet.

### 2.5. Comparisons between ISO 20079 and the New Root-Regrowth Method

Various aspects of ISO 20079 and the new root-regrowth method were compared (Table 1). The ISO test involved dispensing 100 mL of control (Steinberg medium only) and test solutions into 65 mm diameter beakers. Ten to sixteen *L. minor* fronds (two or three fronds per colony) were placed in each beaker, which were then covered and sealed with sealing tape to prevent evaporation before being incubated at 24 ± 2 °C. For comparisons, 3,5-dichlorophenol was used as a reference toxicant, as recommended in the ISO 20079 protocol [22]. After 7 d, all plants were carefully harvested using plastic tweezers and their frond numbers, dry weights and chlorophyll contents quantified. Dry weight was measured by drying the collected fronds at 60 °C until a constant weight was achieved, while chlorophyll concentrations were determined by spectrophotometry after being extracted in 95% ethanol, and quantified using the equation recommended in the ISO protocol [22]. The root-regrowth method was performed as described above.

### 2.6. Inter-Laboratory Comparison Test

An interlaboratory comparison of the *L. minor* root-regrowth test was carried out by 10 international organizations with Cu as the standard toxicant and 5 national organizations with wastewater. Validation was performed using the following steps: First, control charts were created using all control data recorded by the participant laboratories to measure laboratory precision and to monitor culture health. Second, both Grubbs’ test and Dickson Q test were performed to determine whether there were outliers in the data set before collecting and calculating the mean and 95% CIs of the control root lengths from all participating laboratories.

Third, a graphical representation of the statistical analysis (ISO 5725-2 2002) was constructed to observe the distribution of data and to estimate the ratio between the repeatability and reproducibility values. Fourth, another graphical representation of the warning chart approach was constructed to determine the total number of tests that fulfilled the acceptance criteria. Finally, all data sets, including all EC_50_ values; no outliers were used to calculate the final acceptance range of EC_50_ values.

### 2.7. Statistical Analysis

EC_x_ values (effective concentrations at which x% inhibition occurs), with 95% confidence intervals, were calculated from the test results using the linear interpolation method (ToxCalc 5.0, Tidepool Science, McKinleyville, CA, USA). The major and combined effects were tested to confirm the significant differences in the root-regrowth response via analysis of variance (ANOVA). The differences between factor levels were further analyzed using the least significant difference (LSD) test at *p* < 0.05. The coefficient of variation (CV), i.e., the standard deviation expressed as a percentage of the mean, was calculated to estimate the precision of the tests.

## 3. Results and Discussion

### 3.1. Comparisons of the ISO 20079 Protocol vs. the Root-Regrowth Test

We compared various aspects of ISO 20079 and the new root-regrowth method (Table 2).

The EC_50_ (2.44 mg L^–1^) calculated using the new *L. minor* root-regrowth method fell between the EC_50_ values calculated based on dry weight (2.25 mg L^–1^) and frond numbers (3.51 mg L^–1^) using the ISO 20079 test (Table 2). These results indicate that the sensitivity and precision of our new test are similar to or greater than those of the ISO standard *L. minor* testing method.

### 3.2. Interlaboratory Comparison

We conducted an interlaboratory comparison of the results from 10 international laboratories with experience in toxicity assays to validate the *L. minor* root-regrowth test method and to determine the uncertainty of the results (Table 3 and Table 4).

The Dixon test is easy to perform and is mainly applied to small data sets. This test is based on a comparison of the difference between the suspect value and its direct or close neighbor with the overall range or a modified range. It suffers, however, from the masking effect if multiple outliers are present. Therefore, the technical committee of ISO replaced the Dixon test with Grubbs’ test. Grubbs’ test provides different outlier tests for single outliers (highest or lowest value) and for double outliers (two highest or two lowest values). As a modification of the latter, the so-called paired test for two outliers, which can either be situated at the same or different ends of the ordered data, has been proposed. The single outlier test is often combined with one of the tests to detect two outliers.

When both Grubbs’ test and Dixon Q test were performed, there was no significant difference (at the *p* = 0.01 and 0.05 levels) in either test for rejecting a null hypothesis (H_0_: there are no outliers in the data set). Grubbs’ test showed that all Z-scores of the control values of the 10 laboratories were smaller than the critical Z-scores, Z critical = 2.28995 at *p* = 0.05 and Z critical = 2.48208 at *p* = 0.01, indicating that the null hypothesis was accepted. After applying the Dickson Q test, the calculated Q value (0.086) was smaller than the Q critical values (Q critical = 0.466 at *p* = 0.05 and Q critical = 0.568 at *p* = 0.01), indicating that the null hypothesis was accepted.

In the interlaboratory comparison, the mean and 95% CIs of the control root length values were 28.93 ± 20.28 mm (Table 3, Figure 3A). Therefore, the experiment should be repeated if the root length of the control is outside the mean determined by interlaboratory tests or if the coefficient of variation upon repetition of the same concentration is above 30% [20].

Table 5 shows a statistical summary of the interlaboratory test results from the *Lemna* toxicity test. ‘Repeatability’ refers to the closeness of agreement of values when variance between values was estimated within a laboratory during the shortest practical time period by a single operator with a single system from the same test material under identical test conditions. In contrast to repeatability, ‘reproducibility’ describes the variability between single test results obtained from the same sample in multiple laboratories. International standardization agents set an allowable range of repeatability and reproducibility, which is less than 30%. We also created graphical representations of the statistical analysis and the warning chart approach.

As shown in Figure 4B, there were no outliers of EC_50_ values among the 10 laboratories. If the ratio between the reproducibility standard deviation (S_R_) and repeatability standard deviation (S_r_) is below 4, the test method can be considered fairly robust. The S_R_ and S_r_ values were 0.0915 and 0.0715, respectively. In this respect, the current *Lemna* bioassay is a robust test method.

Warning limits and corresponding warning labels are normally used to interpret the results of tests with reference toxicants in order to assess changes in the sensitivity of the organism and the precision within the laboratory. As shown in Figure 3B, there were no test results outside the warning limits, which indicates that there were no outliers when calculating the total mean and standard deviation as well as S_R_ and S_r_.

Therefore, we conclude that the information presented in Table 6, which provides a summary of interlaboratory comparison test results from the *Lemna* bioassay on aquatic toxicity, is valid and reliable. For internal quality control, reference tests using CuSO_4_ (CAS No. 7758-99-8) as a reference chemical should be run periodically to determine whether the *L. minor* being tested responds to a known chemical in the expected manner. Assuming that all recommended procedures and conditions are followed, the mean root length of the controls and mean EC_50_ (±95% Cl) should be 27.22 ± 12.06 mm and 0.335 ± 0.074 mg L^−1^, respectively.

The results of the interlaboratory test for wastewater were performed in Table 7 and Table 8. Figure 5A, B show that there were no outliers in the EC_50_ values (%) of wastewater toxicity tests from the five national laboratories. When we performed both Grubbs’ test and Dixon’s Q test to identify outliers, there was no significant difference (at *p* = 0.01 and 0.05) in either test to reject the null hypothesis (H_0_: there are no outliers in the data set). Grubbs’ test showed that the calculated G-score (1.550) was lower than the critical G-scores (G_critical_ = 1.67 at *p* = 0.05 and G_critical_ = 1.75 at *p* = 0.01), indicating that the null hypothesis was accepted.

After applying Dixon’s Q test, the calculated Q value (0.542) was lower than the Q critical values (Q_critical_ = 0.71 at *p* = 0.05 and Q_critical_ = 0.821 at *p* = 0.01), indicating that the null hypothesis was accepted.

## 4. Conclusions

The root regrowth bioassay differs in several key aspects from three internationally standardized methods: (a) the test can be completed within 72 h; (b) the test vessel is a 24-well plate; (c) the required volume of test water samples is only 3 mL and (d) roots are excised prior to exposure, with subsequent measurements done on newly developed roots. Excising roots prior to exposure precludes the requirement to preselect roots of uniform length, which reduces the need to handle these fragile roots. The artificial severance of roots may never occur in natural settings, as root abscission in *Lemna* has not previously been reported; however, a recent study showed that the tiny, globally distributed water ferns of the genus *Azolla* lose their roots under stressful conditions [31]. This phenomenon, known as rapid root abscission, is believed to free fronds from root-entangled mats and facilitate their dispersal into potentially better environments, representing an important survival strategy for *Azolla*. This observation suggests that the endpoint of *L. minor* root regrowth could have ecological relevance, although further research is needed to verify this hypothesis.

A good biological toxicity technique should be quick and simple to use, while still maintaining sensitivity to the toxicants. In this respect, our 3-d test may be considered a modified version of the 7-d standardized frond test using *Lemna*. Testing time is an important factor in the selection of an appropriate bioassay, as management decisions should be made in a timely manner in cases of unpredictable pollution events.

In interlaboratory tests with ten international and five national participating laboratories for Cu and wastewater, respectively; the validity criteria, represented by repeatability and reproducibility, were well within the generally accepted levels of <30% to 40%, indicating that the current test method is acceptable as a standardized biological test and can be used as a regulatory tool. The conventional ISO 20079, based on frond growth and biomass of *Lemna minor*, is an important tool for aquatic ecosystem and water quality monitoring and management, especially in countries with well-established effluent control programs. However, we live in an ever-changing world that requires constant improvements to meet all the needs of our society, including less costly monitoring tools. In this respect, the *Lemna* root growth test is such an innovation.

Moreover, a single bioassay can never provide a complete picture of environmental quality, as no single test is universally sensitive to all pollutants. In ecotoxicology, representative, cost-effective and quantitative test batteries should be developed to study the effects and mechanisms of action of environmental pollutants. The *Lemna* root growth test complements the traditional *Lemna* standard ISO 20079 and is particularly suitable for rapid screening of wastewater and priority substances spikes in natural waters.

## Figures and Tables

**Figure 1 biology-11-00037-f001:**
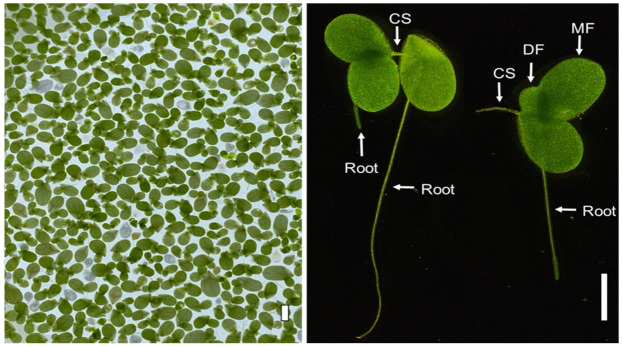
The common duckweed *Lemna minor* 5631. Numerous fronds are shown (**left**), and representative plants are shown in side view (**right**). CS, Connective stalk; DF, daughter frond; MF, mother frond. Scale bar = 2 mm.

**Figure 2 biology-11-00037-f002:**
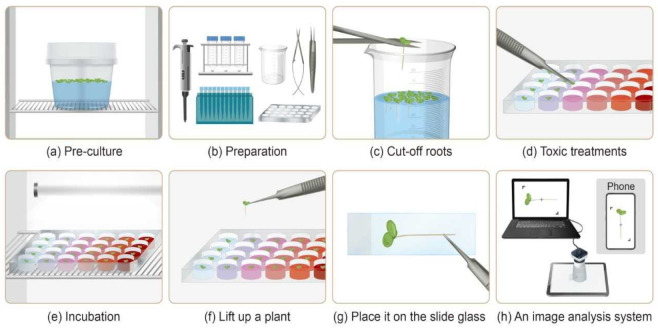
Details of *Lemna* root regrowth test protocol. (**a**) Preculture *Lemna* in a culture vessel. (**b**) Prepare experimental tools for toxicity tests. (**c**) Lift the plants with scissors and cut the root. (**d**) Place one plant without the root per well containing the test solution. (**e**) Cultivate under optimal environmental conditions. (**f**) After 72 h, lift the colony from the surface of the test solution with tweezers and (**g**) Place the colony on a slide glass with the roots aligned. (**h**) Measure the length of the longest roots using an image analyser or mobile phone.

**Figure 3 biology-11-00037-f003:**
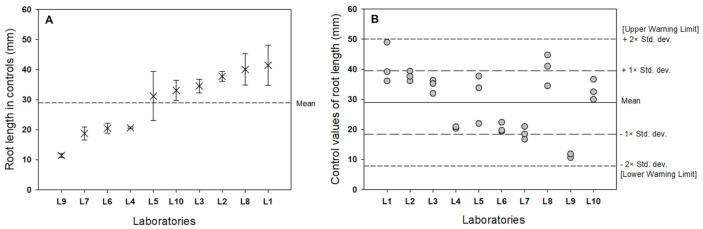
(**A**) Mean root length ± standard deviation of control samples from 10 laboratories. (**B**) Graphical representation of the warning chart approach to the control root lengths of *Lemna minor* from 10 laboratories.

**Figure 4 biology-11-00037-f004:**
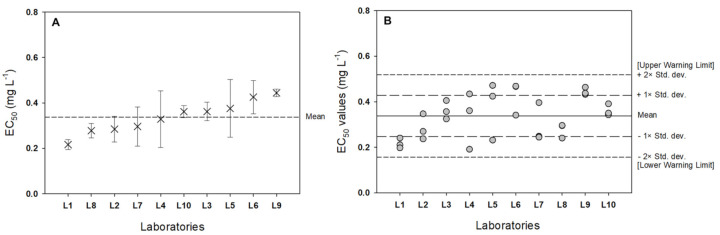
(**A**) Mean regrown EC_50_ ± standard deviation from 10 laboratories. (**B**) Graphical representation of the warning chart approach.

**Figure 5 biology-11-00037-f005:**
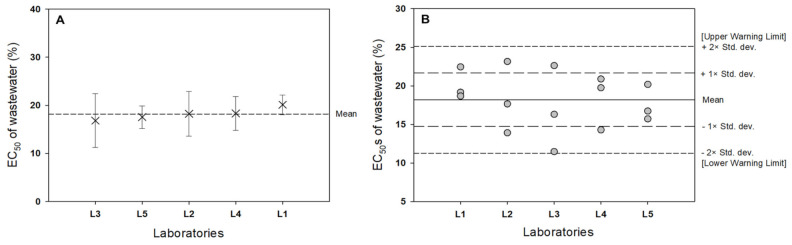
(**A**) Mean EC_50_ ± standard deviation from the interlaboratory wastewater toxicity test. (**B**) Graphical representation of the warning chart approach to the EC_50_ of wastewater from 5 laboratories.

**Table 1 biology-11-00037-t001:** Comparison of the test conditions used for the root-regrowth assay described in the current study and three internationally standardized methods.

Characteristic	This Study	ISO 20079	OECD 221	USEPA
Test species	*Lemna minor* 5631	*Lemna minor* 9441	*Lemna gibba*, *Lemna minor*	*Lemna gibba* G3, *Lemna minor*
Test duration	72 h	168 h	168 h	168 h
Temperature	25 ± 1 °C	24 ± 2 °C	24 ± 2 °C	25 ± 2
Photon Flux Density	90–100 μmol photons m^–2^ s^–1^	85–135 μmol photons m^–2^ s^–1^	6500–10,000 lux	4200–6700 lux
Photoperiod	Continuous light	Continuous light	Continuous light	Continuous light
Test vessel type	24-well plates	Beaker	Flask, Petri dish	Beaker, flask
Medium	Steinberg medium	Steinberg medium	Swedish Standard (SIS) *Lemna* medium (for *L. minor*) or 20× AAP growth medium (for *L. gibba*)	M-Hoagland’s medium or 20×-AAP nutrient medium
Test solution volume	3.0 mL	100 mL (minimum)	100 mL (minimum)	150 mL
Test solution pH	6.9 ± 0.2	5.5	6.5 ± 0.2	7.5 ± 0.1
Test organism size	One colony per test vessel (two or three fronds per colony)	10–16 fronds per test vessel (two or three fronds per colony)	9–12 fronds per test vessel	12–16 fronds per test vessel
Endpoint	Root regrowth length	Growth rate (frond number, frond area, dry weight, chlorophyll contents)	Average specific growth rate, final biomass, area under the growth curve	Total frond number, growth rate (number of fronds per day), mortality (% of dead fronds to total number of fronds) and dry weight, chlorophyll and pheophytin pigment analyses
Test type	Static non-renewal	Static non-renewal	Static none-renewal	Static none-renewal
Condition	Axenic or non-axenic culture	Axenic culture	Axenic culture	Axenic culture

**Table 2 biology-11-00037-t002:** EC_50_ (mg L^−1^) values (95% CI) and Coefficient of variation (CV, %) of *Lemna minor* derived from the root-regrowth method and ISO 20079 after exposure to 3,5-dichlorophenol.

Methods	Endpoints	EC_50_ (95% CI)	CV (%)
Conventional method	Frond number	3.514 (2.986–3.670)	3.22
	Dry weight	2.250 (0.586–3.187)	20.10
	Chlorophyll *a*	3.349 (3.141–3.520)	1.88
	Chlorophyll *b*	3.425 (3.042–3.639)	2.75
	Carotenoids	3.338 (2.988–3.594)	2.83
This study	Root regrowth length	2.441 (1.239–2.992)	17.31

**Table 3 biology-11-00037-t003:** Interlaboratory precision of control values of root length in *Lemna minor*.

Laboratory	Control Root Length (mm) (95% CI)	CV (%)
Lab 1	41.442 ± 7.596	16.20
Lab 2	37.745 ± 1.815	4.26
Lab 3	34.525 ± 2.568	6.57
Lab 4	20.605 ± 0.302	1.30
Lab 5	31.192 ± 9.291	26.32
Lab 6	20.483 ± 1.889	8.15
Lab 7	18.742 ± 2.402	11.33
Lab 8	40.083 ± 5.869	12.94
Lab 9	11.410 ± 0.760	5.89
Lab 10	33.056 ± 3.811	10.19
**95% PI ***	**28.928 ± 20.280**	

* PI; Prediction interval = Overall Mean ± z × s. **Lab 1**, Incheon National University (Korea); **Lab 2**, Korea Coast Guard Metropolitan Police (Korea); **Lab 3,** National Institute of Environmental Research (Korea); **Lab 4**, Institute of Public Health and Environment Research (Korea); **Lab 5**, Ghent University (Belgium); **Lab 6**, Shanghai Ocean University (China); **Lab 7**, University of Connecticut (USA); **Lab 8**, Friedrich Schiller University Jena (Germany); **Lab 9**, Norwegian Institute for Water Research (Norway); **Lab 10**, Chinese Academy of Sciences (China).

**Table 4 biology-11-00037-t004:** Interlaboratory precision of EC_50_ (mg L^−1^) from the *Lemna* toxicity test.

Laboratory	EC_50_ (95% CI)	CV (%)
Lab 1	0.216 ± 0.025	10.23
Lab 2	0.285 ± 0.064	19.80
Lab 3	0.362 ± 0.046	11.25
Lab 4	0.329 ± 0.141	37.88
Lab 5	0.376 ± 0.144	33.82
Lab 6	0.426 ± 0.083	17.20
Lab 7	0.296 ± 0.098	29.14
Lab 8	0.277 ± 0.036	11.46
Lab 9	0.455 ± 0.019	3.78
Lab 10	0.361 ± 0.030	7.26
**95% PI ***	**0.337 ± 0.138**	

* PI; Prediction interval = Overall Mean ± z × s. **Lab 1**, Incheon National University (Korea); **Lab 2**, Korea Coast Guard Metropolitan Police (Korea); **Lab 3**, National Institute of Environmental Research (Korea); **Lab 4**, Institute of Public Health and Environment Research (Korea); **Lab 5**, Ghent University (Belgium); **Lab 6**, Shanghai Ocean University (China); **Lab 7**, University of Connecticut (USA); **Lab 8**, Friedrich Schiller University Jena (Germany); **Lab 9**, Norwegian Institute for Water Research (Norway); **Lab 10**, Chinese Academy of Sciences (China).

**Table 5 biology-11-00037-t005:** Summary of interlaboratory test results based on control root length (mm) of *Lemna* toxicity test.

Sample	l	n	ο%	X	R(S_R_)	CV-R%	r(S_r_)	CV-r%
Control	10	10	0	28.928	30.127 (10.869)	37.573	11.302 (4.077)	14.095

**l**: number of laboratories after outlier rejection. **n**: number of individual tests after outlier rejection. **ο**: percentage of outliers. **X**: overall mean of results (without outliers). **R**: reproducibility. **S_R_**: reproducibility standard deviation. **CV-R**: coefficient of variation of reproducibility. **r**: repeatability. **S_r_**: repeatability standard deviation. **CV-r**: coefficient of variation of repeatability. **Outliers**: non-valid and valid data not conforming to the ISO procedure are not included in the calculations.

**Table 6 biology-11-00037-t006:** Summary of interlaboratory test results from the *Lemna* toxicity test.

Sample	l	n	ο%	X	R(S_R_)	CV-R%	r(S_r_)	CV-r%
Cu (mg L^−1^)	10	10	0	0.337	0.255 (0.0918)	27.2	0.200 (0.0720)	21.3

**l**: number of laboratories after outlier rejection. **n**: number of individual tests after outlier rejection. **ο**: percentage of outliers. **X**: overall mean of results (without outliers). **R**: reproducibility. **S_R_**: reproducibility standard deviation. **CV-R**: coefficient of variation of reproducibility. **r**: repeatability. **S_r_**: repeatability standard deviation. **CV-r**: coefficient of variation of repeatability. **Outliers**: non-valid and valid data not conforming to the ISO procedure are not included in the calculations.

**Table 7 biology-11-00037-t007:** Interlaboratory precision of EC_50_ (%) ± 95% CI and PI from the wastewater toxicity test.

Laboratory	EC_50_ ± 95% CI	CV (%)
Lab 1	20.109 ± 2.318	10.19
Lab 2	18.253 ± 5.260	25.47
Lab 3	16.812 ± 6.332	33.28
Lab 4	18.321 ± 3.995	19.27
Lab 5	17.550 ± 2.652	13.36
**95% PI ***	**18.209 ± 2.402**	

* PI; Prediction interval = Overall Mean ± z × s. **Lab 1**, Incheon National University; **Lab 2**, Ghent University Global Campus; **Lab 3**, Environmental Technology Center, Environmental Corporation of Incheon; **Lab 4**, Institute of Public Health and Environment Research; **Lab 5**, National Institute of Environmental Research.

**Table 8 biology-11-00037-t008:** Summary of interlaboratory test results from the *Lemna* wastewater test.

Sample	l	n	ο%	X	R(S_R_)	CV-R%	r(S_r_)	CV-r%
Wastewater	5	5	0	18.209	9.405 (3.393)	18.634	10.741 (3.875)	21.280

**l**: number of laboratories after outlier rejection. **n**: number of individual tests after outlier rejection. **ο**: percentage of outliers. **X**: overall mean of results (without outliers). **R**: reproducibility. **S_R_**: reproducibility standard deviation. **CV-R**: coefficient of variation of reproducibility. **r**: repeatability. **S_r_**: repeatability standard deviation. **CV-r**: coefficient of variation of repeatability. **Outliers**: non-valid and valid data not conforming to the ISO procedure are not included in the calculations.

## Data Availability

All the results found are available in this manuscript.

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
