# Peer review of "Interlaboratory Validation of Toxicity Testing Using the Duckweed Lemna minor Root-Regrowth Test"

_biology, 2021, doi:10.3390/biology11010037_

Round 1
Reviewer 1 Report
This study proposed a revised method for toxicity test using Lemna minor root-regrowth approach. The approach was validated across several laboratories. My comments and/or suggestions to improve the manuscript are:
- There are several studies that use root-regrowth or root elongation of other plants for toxicity test. I think some of those studies should be cited, discussed, or compared with the present study.
- Referring to part 2.5 and Table 1, is the solution concentration between the standardized methods and the present study comparable? Furthermore, the pH value of the solution among the studies was different.
- The standard methods consider several parameters for observation, while the present study is just based on the single parameter i.e., root regrowth. I think a discussion on the mechanism of action or toxicity effect on the root growth should be explained or proposed. Can the root growth or root length indicate the toxicity level?
- I would like to suggest the present result to be supported by histological and molecular study.
Author Response
Point by point reply to reviewer’s comments
Manuscript number: biology-1455448
|
Reviewer #1 |
Author’s reply |
|
This study proposed a revised method for toxicity test using Lemna minor root-regrowth approach. The approach was validated across several laboratories. My comments and/or suggestions to improve the manuscript are:
1. There are several studies that use root-regrowth or root elongation of other plants for toxicity test. I think some of those studies should be cited, discussed, or compared with the present study. |
Thank you for your suggestion. We have added the following sentences (Lines 352-360):
A recent evaluation of the ecotoxicological significance of root growth as an end-point has revealed that it is sensitive, precise and ecologically significant compared to more traditional endpoints e.g. frond growth or biomass [26,27]. Gopalapillai et al. [27] identified the average root length of L. minor as an optimal endpoint for the biomonitoring of mining wastewater for three reasons: accuracy (toxicological sensitivity to the pollutant), precision (lowest variance) and ecological relevance (direct exposure to metal contaminated wastewater). Subsequently, Park et al. [26] established a well-defined method to measure toxicity-concentration-dependent inhibition of root regrowth in three Lemna species. |
|
2. Referring to part 2.5 and Table 1, is the solution concentration between the standardized methods and the present study comparable? Furthermore, the pH value of the solution among the studies was different. |
In order to compare the sensitivity of the ISO standard and the newly developed root growth method, the test protocols established for the respective methods had to be followed when conducting the tests. The optimum pH conditions for the ISO20079 method and the root growth method were 5.5 and 6.9 ± 0.2, respectively (Table 1). If the solution concentration and pH were the same for both methods, comparison of the two tests would not have been meaningful because the protocols for the test methods were not followed. For further reference, to establish the proposed method, we first determined the optimal environmental conditions for root regrowth, including photon flux density, pH, salinity and temperature. The pH of the test medium can influence the growth of Lemna plants; for example, the pH can affect the bioavailability of metals by changing their speciation and thus their toxicity. We observed that a pH of 7 is ideal for the regrowth of Lemna roots (Park, et al. "A novel bioassay using root re-growth in Lemna." Aquatic Toxicology 140 (2013): 415–424). In addition, we compared the sensitivity of the Lemna root re-growth bioassay at pH 5.5 and pH 6.9, considering ISO 20079 which is performed at pH 5.5, and found no statistically significant difference (P > 0.05) between the tests. Based on these results, we selected pH 6.9 ± 0.2 as the appropriate pH for the toxicity tests, as this is also the same pH range that is measured in the Steinberg medium. |
|
3. The standard methods consider several parameters for observation, while the present study is just based on the single parameter i.e., root regrowth. I think a discussion on the mechanism of action or toxicity effect on the root growth should be explained or proposed. Can the root growth or root length indicate the toxicity level? |
We only wanted to describe here the protocols of the newly developed bioassay method based on root re-growth of L. minor and to validate this test method. For this reason, we have only focused on one endpoint, root re-growth. As the reviewer has correctly indicated, it is impossible to decipher the mechanism of toxic action in root re-growth inhibition unless multiple endpoints have been studied simultaneously, and we cannot say anything here about the underlying toxic mechanism. However, we have published a paper in which we examined several endpoints, including root re-growth, frond growth, pigmentation, and photosynthesis, and found a close relationship between the toxic effects of some herbicides on root re-growth and electron transport rates (Park, et al. "Comparing the acute sensitivity of growth and photosynthetic endpoints in three Lemna species exposed to four herbicides." Environmental Pollution 220 (2017): 818–827). |
|
4. I would like to suggest the present result to be supported by histological and molecular study. |
As mentioned in the Introduction Section, the main objective of this study was to present a detailed protocol for testing the toxicity of freshwater samples with the L. minor root growth bioassay, to explain the standardized analysis of the data and to validate the new test method through inter-laboratory comparison testing with 10 different institutions from 6 different countries. However, we are currently investigating the molecular mechanisms explaining the inhibition of root re-growth and the genes expressed in association with the inhibition of root growth in response to exposure to toxicants. As for the histological study, we are planning to investigate the histology of root re-growth and its inhibition in the near future. |
Reviewer 2 Report
This is an interesting piece of work presented by Park and its collaborators with a novel, inexpensive and fast toxicity screening protocol.
The idea of conducting different testing in multiple labs gives the protocol more reliability and validity.
The protocol is clear and well detailed, the paper contains sufficient data about comparisons made with the other standard protocols to show more the aims and the results of their work.
This work relies on the root regrowth as the only measurable parameter to estimate the toxicity compared to the other methods based on the study performed by (Park et al., 2017). Did the authors try to evaluate other parameters, or was the root-regrowth length the only parameter that can be properly measured during the 72h time? How confident are the authors about that?
I also have another concern, Why CuSO4 only was used as a reference toxicant while, for example, 3,5-dichlorophenol is used in the ISO 20079?
R36-40. The ideas presented in this paragraph can be combined in one clear sentence.
R49 I think « for CuSO4» is missing.
Regards
Author Response
Point by point reply to reviewer’s comments
Manuscript number: biology-1455448
|
Reviewer #2 |
Author’s reply |
|
This is an interesting piece of work presented by Park and its collaborators with a novel, inexpensive and fast toxicity screening protocol. The idea of conducting different testing in multiple labs gives the protocol more reliability and validity. The protocol is clear and well detailed, the paper contains sufficient data about comparisons made with the other standard protocols to show more the aims and the results of their work. This work relies on the root regrowth as the only measurable parameter to estimate the toxicity compared to the other methods based on the study performed by (Park et al., 2017). Did the authors try to evaluate other parameters, or was the root-regrowth length the only parameter that can be properly measured during the 72h time? How confident are the authors about that? |
Thank you for your compliments and comments. Yes, we have only focused on the root-regrowth here as the main objective of this study was to present a detailed protocol for testing the toxicity of freshwater samples with the L. minor root growth bioassay, to explain the standardized analysis of the data, and to validate the new test method through inter-laboratory comparison testing with 10 different institutions from 6 different countries. In inter-laboratory tests with participating laboratories for Cu and wastewater, the validity criteria, represented by repeatability and reproducibility, were well within the generally accepted values of < 30 % to 40 %, indicating that the current test method is acceptable as a standardized biological test and can be used as a regulatory tool. Also, when performing the Grubbs test and Dixon Q-test for the control values, there was no significant difference in either test to reject the null hypothesis (H0: there are no outliers in the data set), indicating that the test method is stable and reliably reflects the degree of toxicity. |
|
I also have another concern, Why CuSO4 only was used as a reference toxicant while, for example, 3,5-dichlorophenol is used in the ISO 20079? |
Copper sulphate (CuSO4) is an appropriate reference chemical, as it is stable in freshwater and produced meaningful data in the preliminary ring test. Copper has the highest binding affinity for organic matter and is 100% soluble. Therefore, CuSO4 is the best reference chemical, probably much better than 3,5-dichlorophenol and potassium chloride (KCl). However, 3,5-dichlorophenol and KCl could also be used as reference toxicants, as they have been successfully used as reference substances for Lemna testing in other standards and methods. The users may decide which reference toxicant to use in their laboratory. |
|
R36-40. The ideas presented in this paragraph can be combined in one clear sentence. |
We have revised the following sentences (Lines 36-39):
The root growth test is therefore a valuable tool for rapid toxicity screening of wastewater effluents and hazardous pollutants in natural waters because it is simple to perform, quick to conduct, cost-effective to operate, and can have an operational benefits for testing time, since management decisions need to be made promptly in the event of unpredictable pollution events. |
|
R49 I think « for CuSO4» is missing. |
We would like to keep the line because we used the reference material (3,5-dichlorophenol) proposed in the ISO standard for the comparison experiments with the conventional method (ISO20079). CuSO4 is our proposed reference toxicant for the newly developed method. |
Reviewer 3 Report
The manuscript is an interesting study. The experimental design is comprehensive and the data obtained are credible and sufficient. I still have a few questions.
- Moving the roots also cause damage to lemna plants, including affecting their growth and also producing stress response. Lemna fronds are very fragile therefore improper operation easily cause wounds on the fronds. Therefore the plants are not at their best conditions in response to stresses from these contaminants, which will finally affect the test results.
- We cannot guarantee that all roots are completely removed. And the growth of each plant is quite different, more repetitions should be added in practical operation to ensure the accuracy of the test.
- Whether 3 days of regrowth is enough for the measurement of roots? How long can roots grow every day in regrowth condition? Did you compare regrowth of lemna roots in the control group and the test groups from 0d to 7d or longer time? This is the basis for this study. The author can submit it as supplemental
- How will the authors promote their method or apply to practice? Almost all duckweed toxicity tests are conducted according to ISO, OECD and other guidelines at present.
- Line132-134, the study in the reference (20) was conducted by the author in 2017. It’s better to detail describe the sensitivity of the roots to contaminants so that it will be convenient for readers to fully understand.
Author Response
Point by point reply to reviewer’s comments
Manuscript number: biology-1455448
|
Reviewer #3 |
Author’s reply |
|
The manuscript is an interesting study. The experimental design is comprehensive and the data obtained are credible and sufficient. I still have a few questions.
1. Moving the roots also cause damage to lemna plants, including affecting their growth and also producing stress response. Lemna fronds are very fragile therefore improper operation easily cause wounds on the fronds. Therefore the plants are not at their best conditions in response to stresses from these contaminants, which will finally affect the test results. |
Thank you for your comments. Lemna roots are important for plant anchorage, nutrient absorption, and cytokinin biosynthesis but the manipulation of roots by simple severance can be completed easily and does not justify the conclusion that removal of roots prior to ecotoxicological testing is inappropriate. Root excision does not cause any marked physiological disturbance, as determined through frond, root growth, and photosynthesis, nor does it add ecotoxicological stress, as shown in the control chart as seen in Fig. 3. In the root re-growth testing standard, roots are excised prior to exposure to the toxicant with subsequent measurements taken from newly developed roots. Excising roots prior to exposure ensures that there is no need to pre-select roots of uniform length, thereby reducing the duration of handling fragile roots. |
|
2. We cannot guarantee that all roots are completely removed. And the growth of each plant is quite different, more repetitions should be added in practical operation to ensure the accuracy of the test. |
To estimate interlaboratory precision and for monitoring culture health, we performed Grubb’s and Dixon’s Q tests to identify outliers; however, as shown in Fig. 4, there was no significant statistical difference in either test for the rejection of the null hypothesis. Therefore, there were no outliers, indicating that the test results are all valid. The interlaboratory CV-R and CV-r for the individual laboratory control values were 37.57% and 14.1% which are well within the generally accepted criteria for CVR (< 30% to 40%), indicating that the current test method is acceptable as a standardized biological test. For these reasons, the ability to cut roots does not seem to have any influence on the length of the plant's root growth. |
|
3. Whether 3 days of regrowth is enough for the measurement of roots? How long can roots grow every day in regrowth condition? Did you compare regrowth of lemna roots in the control group and the test groups from 0d to 7d or longer time? This is the basis for this study. The author can submit it as supplemental |
We have added the following sentences as a supplementary information (Lines 665-672):
Determining the exposure time has been described by Park [30]. The length of the exposure period is equally important when assessing any toxic effects. Exposure durations from 2 to 7-d were tested using the lengths of regrown roots as the endpoint. In general, sensitivity decreased as exposure time increased, but there was an overlap in the 95% CIs between the 3-d and 6-d exposure period. Since rapid response times are often required to deal with chemical pollution events a short duration period was con-sidered desirable and therefore, based on our findings, a 3-d fixed time period for the L. minor root regrowth test was chosen. |
|
4. How will the authors promote their method or apply to practice? Almost all duckweed toxicity tests are conducted according to ISO, OECD and other guidelines at present. |
The ISO standards publishes standards that guide, provide strategic direction and facilitate all elements of society, including local and national governments, business, industry, and individuals, to succeed in an environment where constant change is required for continuous improvement and innovation. Therefore, these standards must be dynamic and strategic directions must be adapted as necessary. We agree that ISO 20079 is an essential tool for the monitoring and management of the aquatic ecosystem and water quality, especially in countries with well-established wastewater control programs. However, we live in a dynamic world, requiring constant innovations to meet all the needs of our society, including less expensive monitoring tools. In this respect, the Lemna root re-growth test is an innovation. The proposed root re-growth test differs from the internationally standardized (ISO, OECD, and US EPA) methods in several key aspects: a) The test can be completed within 72 h. b) The test is performed in a 24-well cell culture plate. c) The required volume of test sample is 3 ml. d) Axenic cultures, which are costly and difficult to maintain, are not required. e) Roots are excised prior to exposure, thereby eliminating handling of fragile roots to obtain a uniform length. A single bioassay can never provide a complete picture of environmental quality, as no single test is universally sensitive to all pollutants. In ecotoxicology, representative, cost-effective, and quantitative test batteries should be derived to investigate the effects and mechanisms of action of environmental contaminants. The Lemna root re-growth test is useful for the rapid screening of either wastewater effluents or priority substances spiked in water. This method complements the longer ISO 20079 Lemna standard. An advertisement for the new method is not the aim of the manuscript. Rather, we would like to show the advantages of this method and provide reliable protocols for evaluation. |
|
5. Line132-134, the study in the reference (20) was conducted by the author in 2017. It’s better to detail describe the sensitivity of the roots to contaminants so that it will be convenient for readers to fully understand. |
We have added the following sentences (Lines 352-360):
A recent evaluation of the ecotoxicological significance of root growth as an end-point has revealed that it is sensitive, precise and ecologically significant compared to more traditional endpoints e.g. frond growth or biomass [26,27]. Gopalapillai et al. [27] identified the average root length of L. minor as an optimal endpoint for the biomonitoring of mining wastewater for three reasons: accuracy (toxicological sensitivity to the pollutant), precision (lowest variance) and ecological relevance (direct exposure to metal contaminated wastewater). Subsequently, Park et al. [26] established a well-defined method to measure toxicity-concentration-dependent inhibition of root regrowth in three Lemna species. |
Round 2
Reviewer 1 Report
-